# Seasonal and Zonal Succession of Bacterial Communities in North Sea Salt Marsh Sediments

**DOI:** 10.3390/microorganisms10050859

**Published:** 2022-04-21

**Authors:** Dennis Alexander Tebbe, Simone Geihser, Bernd Wemheuer, Rolf Daniel, Hendrik Schäfer, Bert Engelen

**Affiliations:** 1Institute for Chemistry and Biology of the Marine Environment, University of Oldenburg, Carl-von-Ossietzky-Straße 9-11, 26133 Oldenburg, Germany; dennis.tebbe@uol.de; 2Institute of Microbiology and Genetics, Department of Genomic and Applied Microbiology, Georg-August University of Göttingen, 37077 Göttingen, Germany; s.geihser@web.de (S.G.); bernd.wemheuer@biologie.uni-goettingen.de (B.W.); rdaniel@gwdg.de (R.D.); 3School of Life Sciences, University of Warwick, Gibbet Hill Road, Coventry CV4 7AL, UK; h.schaefer@warwick.ac.uk

**Keywords:** microbial communities, intertidal, marine, soil, zonation, seasonality, amplicon sequencing, tax4fun2, gene prediction, nitrogen cycle

## Abstract

Benthic microbial communities of intertidal zones perform important biogeochemical processes and provide accessible nutrients for higher organisms. To unravel the ecosystem services of salt marsh microbial communities, we analyzed bacterial diversity and metabolic potential along the land–sea transition zone on seasonal scales on the German North Sea Island of Spiekeroog. Analysis of bacterial community was based on amplicon sequencing of 16S rRNA genes and –transcripts. Insights into potential community function were obtained by applying the gene prediction tool tax4fun2. We found that spatial variation of community composition was greater than seasonal variations. Alphaproteobacteria (15%), Gammaproteobacteria (17%) and Planctomycetes (11%) were the most abundant phyla across all samples. Differences between the DNA-based resident and RNA-based active communities were most pronounced within the Planctomycetes (17% and 5%) and Cyanobacteriia (3% and 12%). Seasonal differences were seen in higher abundance of Gammaproteobacteria in March 2015 (25%) and a cyanobacterial summer bloom, accounting for up to 70% of the active community. Taxonomy-based prediction of function showed increasing potentials for nitrification, assimilatory nitrate and sulfate reduction from sea to land, while the denitrification and dissimilatory sulfate reduction increased towards the sea. In conclusion, seasonal differences mainly occurred by blooming of individual taxa, while the overall community composition strongly corresponded to locations. Shifts in their metabolism could drive the salt marsh’s function, e.g., as a potential nitrogen sink.

## 1. Introduction

Salt marshes (SM) are among Earth’s most productive ecosystems. Sequestering large amounts of organic carbon per unit area, they have a considerable impact on gas exchange with the atmosphere and the exchange of carbon with coastal waters [1]. They provide numerous economic and ecologic services, such as coastal protection, nutrient remineralization and serve as farmland and fishing grounds [2]. Being dynamic intertidal systems of low-energy shorelines, they are located between the mean high and low water lines. Their ecosystem functioning is strongly dependent on nutrient levels, herbivore densities and sea level rise [3,4,5]. As the interface between the terrestrial and the marine environment, they exhibit elevational, vertical and seasonal gradients in, e.g., salinity, redox potential, carbon, sulfur and nitrogen concentrations [6,7,8]. This also leads to a typical zonation in salt marsh’s flora and fauna [9,10,11], which split European salt marshes in the upper SM, lower SM and the pioneer zone [12,13]. Research concerning the ecological cascades in salt marshes has shown that the ecosystem can be controlled by both, top-down and bottom-up mechanisms [4,14,15]. Most of the organic matter that is produced by halophytes is remineralized within the marsh itself [16]. Benthic bacteria are known to provide nutrients to higher trophic levels by organic matter remineralization in soils, marine sediments and wetlands [17,18,19], thereby shaping the productivity and stability of their ecosystem [20].

In eutrophic environments, microbial abundances and degradation of organic matter are usually driven by the availability of electron acceptors [21]. In addition, the bacterial community composition of salt marshes can strongly differ between locations on a regional scale [22]. Although extreme events such as oil spills and storms or high levels of eutrophication have been shown to drastically influence the natural salt marsh plant and microbe communities [23,24], bacterial communities appear to be generally resistant to large-scale compositional shifts [25]. Comparisons of bacterial communities between salt marsh zones of the same region have shown that differences even occur between locations of high and low inundation frequencies [26,27], and between the lower SM and the pioneer zone [28,29]. The prevalent plant community, stage of succession, soil organic matter contents, as well as phosphorous and nitrogen concentrations, have been identified to contribute to the succession of the bacterial communities. However, other factors also contribute to the divergence of bacterial communities and activities. Carbon remineralization rates, for example, have been shown to vary along salinity gradients, with lower rates for salt marshes compared to freshwater sites and diverging rates for the brakish inbetween [18]. Similarly, increasing salt concentrations can also lead to an enrichment of denitrifying and sulfate-reducing bacteria [30,31]. In addition to the spatial niche-based associations, strongly depending on the environmental factors, temporal adaptations have also been observed in studies of a salt marsh chronosequence. The development stage of a salt marsh may, therefore, also affect microbial community composition and must be taken into account to better understand the succession of bacteria in salt marsh sediments [32].

Thus, knowledge about the succession of benthic bacterial communities and their metabolic capabilities can lead to a deeper understanding of their role in the complex ecological cascades of element and nutrient cycling in the salt marsh ecosystem. Furthermore, the multitude of influencing environmental factors on the zonation raises the question if the differences between the zones are of a sharp or gradual nature. Understanding the steady state and naturally occurring fluctuations within the benthic bacterial community of salt marshes is thereby relevant to define the baseline of ecosystem functioning.

Here, we present a seasonal comparison of the sediment bacterial community structures from a salt marsh of the German North Sea Island of Spiekeroog along an intertidal transect. We aim (i) to compare the resident and active bacterial communities, (ii) to identify seasonal shifts and (iii) to assess how bacterial community composition and metabolic potential differs within the land–sea transition zone, with the focus on the interzonal succession.

## 2. Materials and Methods

### 2.1. Sampling and Site Description

The Island of Spiekeroog is located in the German part of the Wadden Sea (Figure 1). The Wadden Sea is an intertidal zone of the North Sea and extends from Denmark to the Netherlands. It is part of the World Heritage list as it contains a rich and diverse flora and fauna. The back barrier reef of Spiekeroog is covered by a salt marsh environment, which was subject of this study. Samples of soil and sediments were taken along a transect from land to sea, spanning 10 different sampling sites (Figure 1, Appendix A). Seasonal sampling was repeated at these locations in late summer (September 2014), spring (March 2015), mid summer (August 2015) and fall (October 2015). At all sites, 2 g of sediment of the top layers (0–3 cm) were collected and immediately mixed with RNAprotect (Qiagen, Hilden, Germany). Samples were initially stored at −20 °C at the Wittbülten Field Station before transport to the laboratory at −20 °C and subsequent storage at −80 °C until further use.

### 2.2. Nucleotide Extraction and 16S rRNA Amplicon Sequencing

DNA and RNA extraction were performed using the MoBio PowerBiofilm DNA extraction kit and the MoBio PowerBiofilm RNA extraction kit, respectively, as recommended by the manufacturer (MO BIO Laboratories, Carlsbad, CA, USA). Removal of residual DNA was performed using TurboDNase according to the instructions of the manufacturer (Applied Biosystems, Darmstadt, Germany). RNA was purified with the RNeasy MiniElute Cleanup kit (Qiagen). DNA removal was confirmed by a control PCR of the 16S rRNA gene, as described below for 16S rRNA gene PCR. RNA was converted into cDNA by employing the Superscript IV reverse transcriptase and a specific primer (5′-CCGTCAATTCMTTTGAGT-′3) as recommended by the manufacturer (Thermo Fisher Scientific, Schwerte, Germany). Subsequently, residual RNA was removed by the addition of 1 µL RNase H (New England Biolabs, Frankfurt am Main, Germany) to each reaction and incubation for 20 min at 37 °C. Obtained DNA and cDNA were stored at −20 °C until further use. Amplification of the 16S rRNA was performed with 16S rRNA gene primers targeting the V3-V4 region (forward primer: S-D-Bact-0341-b-S-17 5′-TCGTCGGCAGCGTCAGATGTGTATAAGAGACAG-CCTACGGGNGGCWGCAG-3′, reverse primer: S-D-Bact-0785-a-A-21 5′-GTCTCGTGGGCTCGGAGATGTGTATAAGAGACAGGACTACHVGGGTA), as described in Klindworth et al. [35] and Herlemann et al. [36], added adapters for library preparation (underlined). PCRs for DNA and cDNA were performed in triplicates as described by Dinter, Geihser, Gube, Daniel and Kuzyakov [28] with each 25 µL volumes, containing 1× Phusion HF buffer (Thermo Fisher Scientific, Waltham, MA, USA), 400 µM dNTPs, 0.4 µM of the respective primers, 75 mM MgCl_2_, 2.5% dimethylsulfoxide and 0.5 U Phusion High Fidelity Hot Start DNA polymerase (Thermo Fisher Scientific). A total of 25 ng of extracted DNA was added for each reaction. Amplification was carried out using the following cycler settings: initial activation at 98 °C for 30 s and 30 cycles of 98 °C for 10 s, 60 °C for 30 s, and 72 °C for 30 s, with a final incubation at 72 °C for 2 min. Pooled PCR products were purified using a GeneRead size selection kit (Qiagen) and quantified with a Quant-iT dsDNA HS assay kit, as well as with a Qubit fluorometer, as recommended by the manufacturer (Invitrogen, Carlsbad, CA, USA). Paired-end sequencing libraries were generated from purified PCR products using the Nextera^®^ XT DNA library preparation kit (Illumina, San Diego, CA, USA). Sequencing was carried out with a MiSeq-System with the MiSeq reagent kit version 3, as recommended by the manufacturer (Illumina).

### 2.3. Processing of 16S rRNA Amplicon Data

Generated 16S rRNA datasets were processed using qiime2-2021.2 [37] as a wrapper for the denoising algorithm DADA2, further described in the pipeline of Yeh et al. [38]. The scripts were adapted from the collection available at: https://github.com/jcmcnch/eASV-pipeline-for-515Y-926R (accessed on 1 November 2019). To increase reproducibility, all steps were performed in a standardized conda environment.

In short: Primer removal and general trimming was carried out using cutadapt [39], allowing a 20% primer sequence mismatch. Subsequently, 16S rRNA gene data were isolated using the bbtools package and SILVA138 [40]. Removal of low-quality ends (quality score < 30) was carried out by cutting the forward and reverse sequences at 230 bp and 210 bp, respectively. For denoising, merging and chimeric removal, the qiime2 dada2 [41] was used. Taxonomic classification was achieved with the qiime2 classify-sklearn plugin and SILVA138 as a reference database using the Genome Taxonomy Database. Sequences assigned to Archaea, chloroplasts and mitochondria were excluded for all subsequent analyses.

### 2.4. Statistical Analyses

All statistical analyses were carried out with R [42]. Alpha diversity measures, rarefication, NMDS, Mantel test and Bray–Curtis dissimilarities were calculated using the vegan package [43]. Sequence counts for alpha diversity were rarefied to the samples with the minimum read count of 5909 sequences per sample using bootstrapping with 999 iterations. Shannon and Simpson indices were transformed to derive true diversity [44]. Statistical differences were then determined using a one-way analysis of variance (ANOVA), followed by a Tukey’s Honest Significant Difference (HSD) test, considering a difference with *p* < 0.01 (ANOVA) and alpha < 0.05 (Tukey’s HSD) between pairs of means of groups as statistically significant. For the bar plots, all groups on order level with an abundance above 1% in the whole dataset or with an abundance above 1% in 5% of the samples are displayed, while the remaining were grouped into “Others”. To reduce the asymmetry of ASV distributions over the transect, log transformation of proportions normalized per sample were used to calculate the Bray–Curtis dissimilarity for non-metric multidimensional scaling (NMDS) analysis. Statistical significance between pairs of groups were tested by a permutational manova (PERMANOVA) using the adonis2 (vegan package) wrapper pairwise.adonis version 0.4 [45]. Distance–decay correlations were determined with a fitting linear model using the Bray–Curtis dissimilarity and the geographical distance between the samples, which had been calculated by the haversine formula. The statistic relationship was tested using a Mantel test with 9999 runs.

### 2.5. Functional Gene Prediction (tax4fun2)

Functional gene prediction was carried out using tax4fun2 [46]. Therefore, ASV reads of the 16S rRNA gens (DNA-based) and transcripts (RNA-based) were rarefied to the minimum read count of 5909 sequences with a 50× bootstrap and blasted (>97% identity) against the default reference database (Ref100NR). Samples with means of <5% that used ASVs were considered not representative and were excluded from further analysis.

## 3. Results

The sequence analysis of the samples taken in September 2014, March 2015, August 2015 and October 2015 along the land–sea transition zone of Spiekeroog Island, resulted in a total of 2,996,473 quality filtered reads, assigned to 36,648 ASVs. The transect spanned from the most terrestrial upper SM (Upp) towards the marine mudflat (Mud1–2), with the lower SM (Low), five pioneer samples (Pio1–5) and one directly from the shoreline in-between (Edge). The numbers for the samples from the mudflat and the pioneer zone represent intrazonal locations numbered in a seaward direction (Figure 1). Resident and active bacterial communities, here represented by their 16S rRNA genes and transcripts, harbored 59% and 41% of the total read counts, respectively.

### 3.1. Bacterial Alpha-Diversity Varies along the Transect

Alpha diversity measures differed significantly between the sites (Appendix A), exhibiting higher values in the more terrestrial (Upp, Low, Pio1–5) than the marine parts of the transect (Edge and Mud1–2). The mean diversity from Pio1 towards Pio3 were higher compared to Edge and Mud1–2. The mean bacterial richness covered ~79% ± 7 of the expected richness, calculated by Michaelis–Menten kinetics. On a seasonal scale, the mean diversity appeared to be the highest in the March sampling campaign; however, these differences were statistically not significant, probably due to the high heterogeneity between the individual sites of the transect. In line with our expectations that the active communities represent a subset of the resident communities, the resident communities showed significantly higher diversity compared to the active communities.

### 3.2. Distance–Decay Analysis Shows Higher Dissimilarity with Distance

Distance–decay analysis showed a positive correlation between the Bray–Curtis dissimilarity of the samples and the geographical distance. The Mantel test suggested a positive correlation with R^2^ = 0.25 and R^2^ = 0.32 (*p* < 0.0001) between distance and dissimilarity of the resident (Figure 2A) and the active bacterial community (Figure 2B), respectively. However, when sub-setting the seasons, the strongest distance–decay correlation was found in the resident community in March and October (R^2^ = 0.61 and R^2^ = 0.65; *p* < 0.0001), while it was the weakest on gene (R^2^ = 0.02; *p* = 0.82) and transcript levels (R^2^ = 0.23; *p* < 0.0005) in September (Appendix A). Some samples even differed strongly on small distances, which is indicated by the large variation in the Bray–Curtis dissimilarity, e.g., ranging from ~0.4 to ~1 at the same location (0 m distance). However, the overall positive correlation shows a gradual succession in the community composition along the transect which increases with distance.

### 3.3. Zonation Outweighs Seasonality Effect on Bacterial Diversity

The NMDS and PERMANOVA analysis for DNA- and RNA-based bacterial community structures shows a significant shift (Appendix A) between the resident and active bacterial communities (R^2^ = 4.8; *p* ≤ 0.001). This is visible along the NMDS2 axis across sampling times and sites (Figure 3). Additionally, a gradual zonation from the most terrestrial sampling point (Upp) to the marine samples (Mud1–2) is displayed. Here, significant differences were observed between Upp vs. Pio 1–5 (R^2^ = 4.7; *p* ≤ 0.01), Upp vs. Edge (R^2^ = 5.1; *p* ≤ 0.01) and Upp vs. Mud 1–2 samples (R^2^ = 8.2; *p* ≤ 0.01). Similar significant differences were found between Low vs. Pio 1–5, Low vs. Edge and Low vs. Mud 1–2 samples (all *p* ≤ 0.01), as well as for Pio 1–5 vs. the more marine sites (Edge, Mud1–2). Seasonal differences are much less prominent with some exceptions (Sep14 vs. March15: R^2^ = 3.2; *p* ≤ 0.006 and Aug15 vs. March15: R^2^ = 2.9; *p* ≤ 0.012). Overall, these differences are outweighed by the gradual succession over the land–sea transition zone. The observed zonation is in accordance with the commonly used vegetation-based description of European salt marshes, splitting them into distinctive zones. Interestingly, our investigation revealed an additional intrazonal gradient of the benthic bacterial community composition within the pioneer zone. Here, the most terrestrial pioneer sample (Pio1) is very similar to the lower SM (Low).

### 3.4. Bacterial Community Composition Shifts Are Driven by Specific Taxa

Taxonomic analysis at order level revealed changes in the abundance of specific taxa over seasons and sampling sites, with stronger effects observed in the active bacterial communities (Figure 4). For this analysis, all groups at order level, with a dataset-wide abundance of >1% or with abundances of >1% in >5% of the samples, are displayed, while the remaining were grouped into “Others”. Across all samples, the five most abundant classes, Alphaproteobacteria (15%), Gammaproteobacteria (17%), Planctomycetes (11%), Cyanobacteriia (7%) and Acidimicrobiia (7%), account for more than half (57%) of the bacterial community. Differences between the active and resident communities (Appendix A) were most pronounced for the Planctomycetes (17% and 5%), Cyanobacteriia (3% and 12%), Bacteroidia (9% and 2%), Polyangia (2% and 9%) and Verrucomicrobiae (8% and 2%). Seasonality was observed in elevated abundances of Gammaproteobacteria in March15 (25%) and Cyanobacteriia in Aug15 (11%) and Oct15 (13%).

Some taxa also exhibited shifts between sites, with the most prominent example of the Cyanobacteriia. They increased towards the mudflat during the summer and fall samples. This trend was more pronounced in the active bacterial community, with abundances of up to 70% (Aug15) and marking a strong Cyanobacteriia summer bloom in the marine part of the transect. Additionally, we observed a slight but steady increase in mean abundances of the Gammaproteobacteria across the transect (Upp: 12% to Mud2: 20%). Although the mean abundances of the total Alphaproteobacteria did not show a clear trend, differences could be observed at the order level (Appendix A), with increasing abundances of the Rhodobacterales towards the marine sites (Upp: 3.5% to Mud2: 18%). A similar trend was observed for the Bacteroidia, which was mainly driven by the Flavobacteriales (Upp: 2.6% to Mud2: 4.9%). On the contrary, Planctomycetes and Polyangia showed increasing abundances towards the more terrestrial samples, with the Polyangia comprising up to 37% of the active community in March15. The Verrucomicrobiae were one of the few major groups which increased in the central part of the transect. Overall, the dataset points towards a systematic response of specific taxa in presence and activity to spatial differences and seasonal variations within the investigated land–sea transition zone.

### 3.5. Functional Gene Prediction Resolves Zonal Variations

To identify potential metabolic functions of the benthic bacterial communities, the gene prediction tool tax4fun2 was applied to translate DNA-based and RNA-based community structures into virtual metagenomes based on their taxonomic identities. On average, 11.1% ± 4.7 of the ASVs per sample had a hit in the curated tax4fun2 database after the removal of nine samples with too low representation (3.3% ± 1 of ASV with a hit), transforming into a total of 8446 KEGG orthologs (KO). A heatmap of z-scores for the different sampling sites was calculated from the relative abundances of the predicted genes for 16S rRNA genes (Figure 5A) and transcripts (Figure 5B), averaged over all seasons. As benthic bacterial communities are major players in the cycling of sulfur and nitrogen species, we concentrated on the representative genes involved in the conversion of respective compounds. Overall, the general trends for the predicted occurrence of metabolic genes along the transect were similar for both the DNA- and RNA-based analyses, showing slight differences, e.g., for denitrification and sulfate reduction.

The predicted metabolic potential of the bacterial communities for nitrogen fixation appeared to be widespread over the entire transect with lowest values for the upper SM. The genetic potential for nitrification (*pmoABC*-*amoABC*), in turn, appears to be highest in the more terrestrial locations, gradually decreasing towards the marine environment. While the closely related particulate methane monooxygenase (*pmoABC*) has the same KEGG classification as the ammonia monooxygenase (*amoABC*), methane oxidation is not supported by our observations. Here, downstream genes involved in this process (*mxaF* and *xoxF*, encoding for methanol dehydrogenases) do not exhibit a similar abundance pattern (data not shown). However, the high z-scores for *hao* gene-encoding for the hydroxylamine oxidase, the subsequent enzyme in the nitrification pathway, match the *amoABC* abundance and, thus, support nitrification.

Ammonium, which potentially is introduced to the system via terrestrial input, e.g., by livestock farming or organic matter remineralization, can be nitrified in the terrestrial part of the salt marshes, where the oxygenation of the soil is usually higher. The produced nitrite and nitrate can then further be denitrified to N_2_ after being transported to mostly anoxic sediments in the more marine or deeper sediment layers. The intensified denitrification in the marine part of the transect is visible in the opposing trend for the predicted denitrification and nitrification genes, most visible in the RNA-based gene prediction. At our study site, the abundance pattern of both genes encoding for the nitrite reductase (*nirS*, *nirK*) show opposing abundance patterns. The genes could potentially be involved in denitrification and annamox. However, due to the similarity in the abundance of the *nirS* with other denitrification genes (*nosZ*, *norCB*), it is more likely to be involved in denitrification. Predicted gene abundances of dissimilatory nitrate reduction to ammonium (DNRA) did not show a clear trend. Some functional genes were highly abundant in the more terrestrial (*nirB*, *nrfAH*), and others in the marine part (*napAB*, *narIV*, *nirD*). Assimilatory nitrate reduction (ANR) increased towards the terrestrial sites, binding nitrate into biomass. This process might be fed by nitrate from nitrification, leading to an overall removal of ammonia from the environment.

Interestingly, the potentials for the genes involved in the activation of sulfate to APS (*cysND*, *cysNC*) of the assimilatory sulfate reduction (ASR) were also higher in the more terrestrial part of the transect, indicating that these soluble nutrients are incorporated into biomass. Potentials for dissimilatory sulfate reduction (DSR) increased towards the marine sites. Interestingly, this trend is more visible in the DNA-based gene prediction, pointing towards a metabolic potential which was less reflected in the RNA-based analysis.

## 4. Discussion

In this study, we have investigated how the composition and function of bacterial communities shift along an elevational transect from more terrestrial to marine salt marsh sediments over a seasonal period. Our aim was to investigate these variations to understand to what extent the different environmental conditions drive differences in bacterial community structure and function. We demonstrate that the bacterial community exhibits a zonation with gradual inter- and intrazonal successions. While salt marsh zonation is usually defined by inundation frequencies and specific plant associations, our findings deepen the understanding of the commonly used terminology by a gradual intrazonal succession of the benthic bacterial community.

### 4.1. Composition of Salt Marsh Mircrobial Communities Changes on Spatial and Seasonal Scales 

The salt marsh of the back barrier reef of the North Sea Island of Spiekeroog was mainly dominated by Alpha- and Gammaproteobacteria. Our finding that the community composition changes along the land–sea transition zone is in accordance with previous studies on salt marsh microbial communities from the North Sea Islands of Spiekeroog and Schiermonnikoog. While the abundance patterns found in our study differ for some taxa, zonal successions between the upper and lower SM [26] and between the lower SM and the pioneer zone were detected as well [28]. A study of a salt marsh chronosequence revealed that a bacterial community composition is strongly associated with the successional stages of the salt marsh and also correlates with short-term seasonal changes in environmental parameters [32]. In contrast, far-distant salt marshes showed an even stronger divergence. For instance, Angermeyer, Crosby and Huber [25] investigated salt marshes from the U.S. east coast. The sampling sites were chosen in proximity to *Spartina alterniflora* species. They showed that some locations were dominated by Gammaproteobacteria and other sites by Chloroflexi and Deltaproteobacteria. While Gammaproteobacteria were also among the most abundant groups in our marsh system, their abundances at the southern U.S coast were much higher (up to 50%) compared to the *Spartina*-dominated sampling sites in our transect (16.7% ± 1.2, Pio1–5). However, it must be noticed that the distance should not be understood as an explanation for the above-mentioned differences in community compositions, but it hints towards the understanding of locally adapted bacterial communities to the distinct conditions in their environment.

Chuvochina et al. [47] took a similar approach to our study. They sampled the North Stradbroke Island in Queensland (Australia) along a salinity gradient from freshwater wetlands via salt marshes to mangrove sediments. Here, Alphaproteobacteria were the most abundant in the freshwater environment, whereas the salt marshes showed an increase in Actinobacteria, Chlorobi, Gamma- and Deltaproteobacteria, with the latter showing higher abundances in the mangroves. Similar to our observations, increasing differences with distances between the sampling sites were found in 16S rRNA gene amplicon data [29,48] and at the functional gene level of sulfate-reducing bacteria [22]. Interestingly, while we observed a dynamic system including differences between seasons, the overall community composition of salt marshes was shown to be generally resilient to dispersal and compositional shifts on seasonal scales [22,25]. The detection of the summer bloom of Cyanobacteria is in accordance with investigations on microphytobenthos communities in coastal environments. Barranguet et al. [49] found seasonal changes from diatom-dominated communities in winter and spring, to cyanobacteria-dominated communities in summer. The authors have attributed this shift to specific adaptations to light and temperature conditions, with Cyanobacteria being better adapted to high temperatures [50].

### 4.2. Cyanobacteria and Verrucomicrobiae Exhibit Different Response Strategies to Changing Environmental Conditions 

Interestingly, in our investigation, the shift in the occurrence of Cyanobacteria was much more pronounced in the analysis of 16S rRNA transcripts than that based on 16S rRNA genes. Previous work has shown that some Cyanobacteria growth rates and rRNA contents are positively correlated, and thus, the increased rRNA of Cyanobacteria in our samples is likely reflecting their higher growth rates and activity during the summer [51]. Even though it is debatable that active bacteria produce a higher number of ribosomes, and thus, of ribosomal RNA, the comparison of 16S rRNA genes and transcripts as indicators for the resident and active bacterial communities, respectively, is frequently employed (reviewed in [52]). A contrasting example for different patterns in 16S rRNA gene and -transcript abundance is the Verrucomicrobiae. These organisms did not show a seasonal pattern, but there was a pattern in their spatial distribution with increased abundances in the pioneer zone. However, they appear to be more abundant in the resident than in the active community, which was also observed in a comparison of metagenome and -transcriptome data by White et al. [53]. Thus, these bacteria probably have a different response strategy to changing environmental conditions.

### 4.3. Inter- and Intrazonal Successions of the Bacterial Communities along the Salt Marsh Transect 

Our study and other investigations mentioned above have shown that benthic bacterial communities can differ seasonally and interzonally. Furthermore, our distance decay and community analyses have shown a gradual intrazonal shift in the bacterial composition. Small-scale intrazonal community shifts were already demonstrated in the early study by Franklin et al. [54], which performed a polymorphic DNA fingerprinting analysis on a narrow transect across a tidal creek. While the variance in their transect was strongly influenced by the elevation, our investigations show a strong dependency on the distance between sampling sites. However, this is not contradictory, taking into consideration that our transect mostly follows an elevational gradient from Upp with 1.48 m above mean sea level (MAMSL) to Mud2 with 0.945 MAMSL (Appendix A). Taken together, this supports our idea of small-scale intrazonal community shifts in dependence on the distance between sampling sites and their elevation, leading to locally adapted bacterial communities in salt marsh sediments across the land–sea transition zone. While some of the zonation-relevant variables are well described and similar between distant locations, other factors are still not fully understood [12,55,56]. However, the co-occurrence of certain bacterial taxa and the specific vegetation of salt marshes hints towards a tight interaction [26,57,58]. To what extent a bacterial community composition is a result of the environmental conditions or is a shaping variable within the salt marsh environments remains to be elucidated.

### 4.4. Predicted Gene Abundances Show Gradual Changes in Nitrogen and Sulfur Cycling Potentials

The input and cycling of nitrogen in salt marshes has received considerable attention over the past decades [59]. Some authors have observed an increase in aboveground plant growth due to higher nitrogen availability [60,61]. On the other hand, the increased nitrogen availability can be accompanied by a reduced root production, which can facilitate erosion, thereby negatively affecting salt marsh sediment stability [3,62]. While, unfortunately, our observations were not accompanied by measurements of the various nitrogen species, the presented gene prediction indeed indicates the ability of salt marsh benthic bacterial communities to cycle multiple nitrogen species along the land–sea transition zone.

Higher nitrification rates in the more terrestrial salt marsh sediments would potentially reduce the amounts of ammonia at these sites. Additionally, the initial step in nitrification might also be performed by ammonia-oxidizing archaea. Thus, the predicted potential for this process might even be underestimated in our investigation, as our analysis exclusively concentrates on bacteria. Ammonia oxidation would counteract the natural and anthropogenic inputs from land, while also increasing the concentrations in nitrite. However, it must be noticed that the fraction of reactive N-species in watersheds commonly feeding wetlands is increasingly dominated by nitrate when total nitrogen concentrations increase [63]. Interestingly, when nitrate is delivered through flooding, it was shown to have only a limited influence on the growth of salt marsh vegetation [64]. This goes along with our observations that all denitrification genes and the DNRA genes *napAB* and *narIV* are predicted at a higher frequency in the more marine parts of the salt marsh where flooding occurs in higher frequencies. Additionally in accordance with the observation of Bowen et al. [65], the eutrophication of salt marshes by nitrate strongly increases DNRA, where nitrate is used as an electron acceptor for microbial respiration and nitrite is further reduced to ammonia rather than to NO, N_2_O and N_2_. The enhanced bacterial nitrate respiration observed in the present and previous studies, together with the preferential uptake of ammonium over nitrate by macrophytes, can be an explanation to the reduced effects of seaward nitrate intrusion [64,65]. However, these trends might only partially be transferable to other wetlands. Salt intrusion and sulfide enrichments in contrast have been shown to promote DNRA over denitrification, thereby reducing the N_2_ emission to atmosphere in the more sulfidic marine parts of these marshes [30,66].

Nitrification at the terrestrial sites and denitrification in the marine zone of our salt marshes exhibit synergistic effects for nitrogen removal. Together with the ability to reduce oxidized nitrogen species along the whole transect, these processes are underpinning the functioning of wetlands in mitigating reactive nitrogen species of the downstream water bodies [67,68]. While denitrification removes nitrogen from the salt marshes by N_2_ formation, assimilatory nitrate and sulfate reduction increased towards the terrestrial zones. This might lead to a higher availability of organic nitrogen sources in the ecosystem, which have a smaller erosion ratio compared to inorganic nitrogen, reducing losses due to erosion [69]. The bacterial conversion to organic nitrogen- and sulfur-sources can thereby contribute to a more stable system, delivering accessible nutrients when needed. Taken together, our findings might be an indication that assimilatory reduction in oxidized inorganic compounds for biosynthesis plays a larger role in the upper and lower SM compared to the marine part of the transect where the abundance of the reduced forms of N and S are likely more readily available due to the high prevalence of the reductive dissimilatory pathways.

### 4.5. Ecological Considerations

The ability of salt marshes to convert and remove reactive nitrogen species becomes even more important, given the destruction of these habitats by sea-level rise and human exploitation [15,70]. Together with the increasing inputs of N-species from land in the past decades [71], this would further enhance the inflow of nutrients to rivers and oceans. This potential additional inflow of nutrients from land could promote large algal blooms in coastal waterbodies, followed by a complex succession of microbial heterotrophic activity [72], potentially leading to oxygen depletion and a reduction in water quality. Furthermore, it could irreversibly damage the balance of natural oceanic communities, including fish populations, thereby having strong implications for human food availability and nature preservation. Thus, it is not only of great interest to intensify the ongoing efforts to reduce the anthropogenic nitrogen inputs to the environments [73], but also to increase the attempts to protect and restore wetlands [68], including salt marshes. Detailed investigations of the N-cycle, combining N-measurements and conversion rate measurements with the microbial diversity and their metabolic potential, would deepen our understanding of salt marsh functioning.

## 5. Conclusions

Our results suggest a gradual inter- and intrazonal shift in bacterial community structure and function along the land–sea transition zone of salt marshes. The trend was consistent over all seasons and for the resident and active communities. However, seasonal differences appeared to be stronger in the active bacterial fraction, most pronounced in the cyanobacterial summer bloom. Furthermore, the distance decay behavior strengthens the idea of locally adapting bacterial communities in salt marsh sediments. Thus, the general trend of increasing community difference with distance between locations can be extrapolated. Gene prediction analysis showed high potentials for nitrification accompanied by assimilatory processes in the terrestrial zones shifting to denitrification and dissimilatory processes in the more anoxic marine part. This provides new evidence for the idea that salt marshes, and potentially other wetlands, serve as nitrogen sinks, buffering high inputs from land to sea.

## Figures and Tables

**Figure 1 microorganisms-10-00859-f001:**
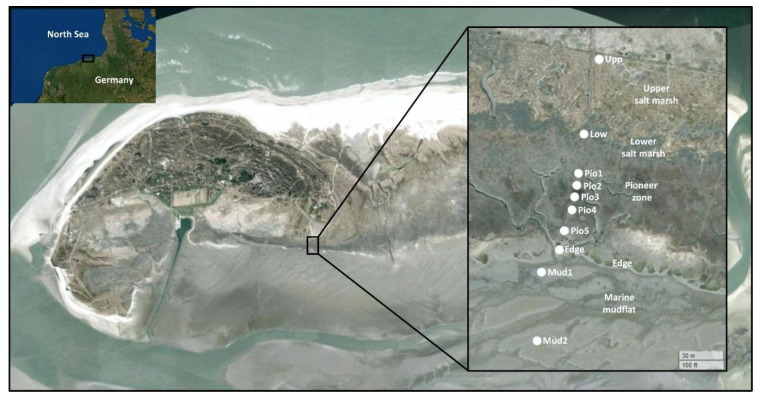
Location of sampling sites on the back barrier reef of the German Wadden Sea Island of Spiekeroog. The sampling sites are designated as “Upp” for the upper SM, “Low” for the lower SM, “Pio1–5” for the five sampling sites from the pioneer zone, “Edge” for the shoreline and “Mud1–2” for the mud-flats. The map was created in R using the leaflet package [33] with World Imagery [34]. Sources: Esri, DigitalGlobe, GeoEye, i-cubed, USDA FSA, USGS, AEX, Getmapping, Aerogrid, IGN, IGP, swisstopo, and the GIS User Community.

**Figure 2 microorganisms-10-00859-f002:**
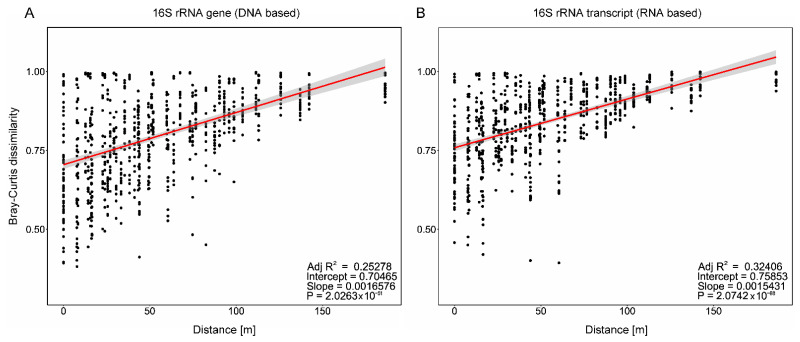
Distance–decay analysis. Pairwise Bray–Curtis dissimilarity over the geographic distance for (**A**) 16S rRNA genes (DNA-based) and (**B**)-transcripts (RNA-based), independent of location and season. The red line represents a fitting linear model, with the 95% confidence level interval as the shaded area.

**Figure 3 microorganisms-10-00859-f003:**
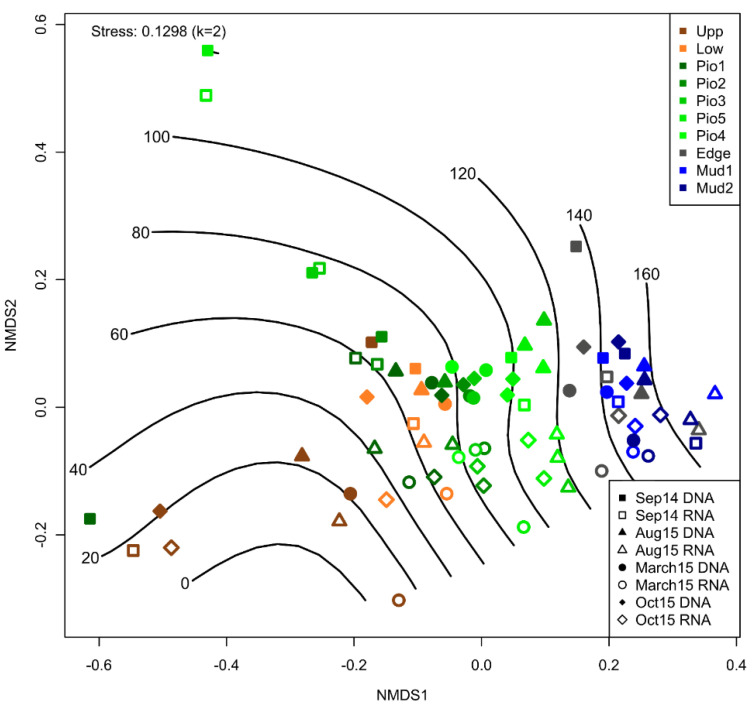
NMDS analysis based on a Bray–Curtis dissimilarity matrix for 16S rRNA genes (filled symbols) and transcripts (open symbols). Colors represent the zones along the transect. Symbols display the different sampling dates. The lines show the fitted geographic distance to the upper salt marsh in meters.

**Figure 4 microorganisms-10-00859-f004:**
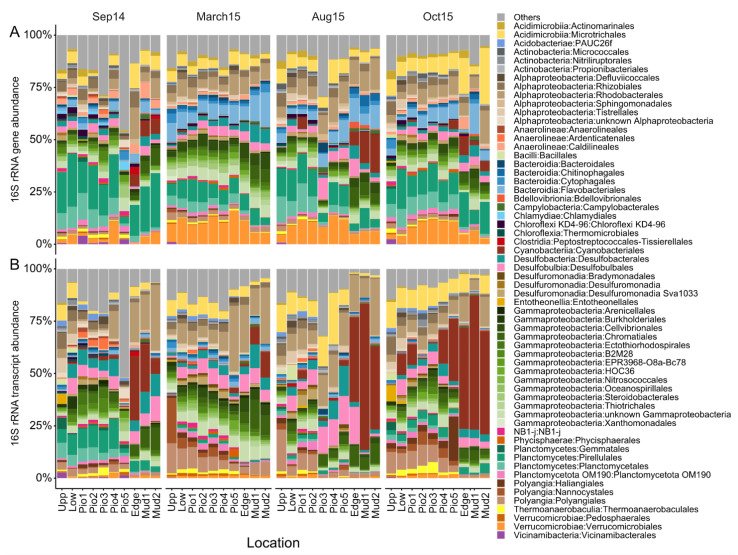
Bacterial community composition. Relative abundance of bacterial (**A**) 16S rRNA genes (DNA-based) and (**B**)-transcripts (RNA-based). ASVs were grouped at the order level. Taxa shown have a dataset-wide abundance of >5% or abundances of >1% in >5% of the samples, the remaining are grouped into “Others”.

**Figure 5 microorganisms-10-00859-f005:**
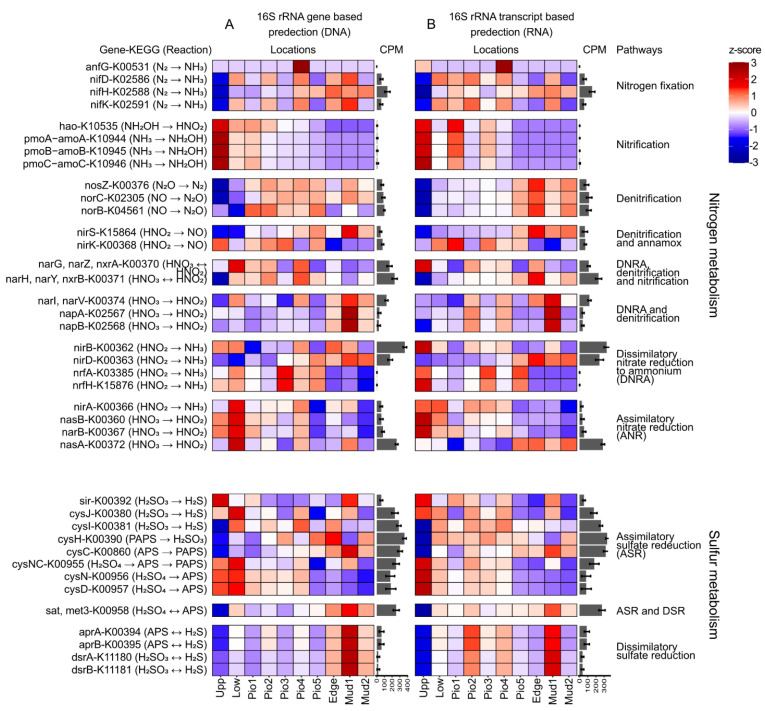
Functional gene prediction. Heatmap of a functional gene prediction (tax4fun2) for (**A**) 16S rRNA genes (DNA-based) and (**B**) 16S rRNA transcripts (RNA-based). Color intensities in the heatmap display the z-score of the mean abundances of predicted KEGG orthologs. Mean counts per million (CPM) are given in a bar chart next to the individual genes.

## Data Availability

Sequence data were deposited in the Sequence Read Archive (SRA) of the National Center for Biotechnology Information (NCBI). Under the accession number “SRP107906”, sequence data of the samples are provided. The used biosamples and meta information are summarized in Appendix A.

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
