# Peer review of "Seasonal and Zonal Succession of Bacterial Communities in North Sea Salt Marsh Sediments"

_microorganisms, 2022, doi:10.3390/microorganisms10050859_

Round 1

Reviewer 1 Report

This study merits in exploring zonation of bacterial diversity and potential functioning using both rDNA and rRNA markers. However, the major flaws lie in two aspects: lacking physiochemical parameters, and not making full use of the advantages of RNA data. The authors have acknowledged that “the active communities represent a subset of the resident communities”. In agreeing so, a mixture of DNA and RNA data for community similarity decay analysis (as shown in Fig. 2) did not make any sense. DNA and RNA data should be analyzed separately (e.g., by showing two fitting lines in Fig. 2). This also applies to functional inference (Fig. 5), and rRNA-based inference is apparently better than the rDNA-based, because relic DNA and DNA extracted from dead, resting or dormant cells included in the DNA pools are not functionally active.   

Line 15: change “their” to “bacterial”

Lines 125-127: Were all samples used the same set of adapters?  Usually, different adapters or barcodes are used for separating data of specific samples in silico.

Lines 177-181: Were functions predicted using DNA or RNA data?

Line 195-196:  “Overall, the mean diversity and richness from Upp towards Pio3 were higher compared to values of Pio4 to Pio5” ---- This is not true. They are not significantly different according to Fig. S1.

Line 199:  change  “was” into  “appeared to be”  

Lines 200-201: “probably due to the high heterogeneity between the individual sites of the transect”

Line 227: “A clear separation of samples from more terrestrial versus marine sites…”  Basically, this hypothesis on difference in community structure should be statistically tested using, for example, ANOSIM or PerMANOVA, rather than simply visual perceives.

Line 240:  “the fitted geographic distance between the samples in meters. “---- This is apparently incorrect.

Figure 4. Please label DNA and RNA to the two figures.

In the section of “3.5. Functional gene prediction resolves zonal variations”, trends of functions are generally described, but not statistically tested, making the conclusions weak.

The discussion should be greatly simplified, as many points only are reviewing previous work (thus do not really matter) rather than focusing on new findings of this work. Also, similar studies and existing knowledges on the tidal zonation effects on bacterial diversity and functions in other plants-dominated coastal ecosystems (e.g., mangrove) may be compared and discussed.

Author Response

We thank reviewer 1 for the constructive comments and hope, we could clarify most of the adressed issues. Please find our answers below.

Comment: This study merits in exploring zonation of bacterial diversity and potential functioning using both rDNA and rRNA markers. However, the major flaws lie in two aspects: lacking physiochemical parameters, and not making full use of the advantages of RNA data.

Answer: We agree, the physiochemical parameters are lacking. The only parameters that were measured during the campaigns were e.g. water content, porewater salinity, temperature or atmospheric pressure. Additionally, these parameters were only measured for a subset of samples and not for all sites along the transect. Thus, we can not present a meaningful set of environmental data relevant for our analysis (e.g. concentrations of nitrate, nitrite, ammonium, sulfate, …). We have now indicated the lack of physiochemical parameters in the discussion.

C: The authors have acknowledged that “the active communities represent a subset of the resident communities”. In agreeing so, a mixture of DNA and RNA data for community similarity decay analysis (as shown in Fig. 2) did not make any sense. DNA and RNA data should be analyzed separately (e.g., by showing two fitting lines in Fig. 2). This also applies to functional inference (Fig. 5), and rRNA-based inference is apparently better than the rDNA-based, because relic DNA and DNA extracted from dead, resting or dormant cells included in the DNA pools are not functionally active.

A: Thank you for the comment. We have now split the distance decay analysis (Figure 2) and the functional inference (Figure 5) into DNA and RNA based analyses and changed the description of the results accordingly. In figure 5, we now present both analyses to provide a full picture to give the reader the opportunity to view both approaches.

C: Line 15: change “their” to “bacterial”

A: Done as suggested

C: Lines 125-127: Were all samples used the same set of adapters?  Usually, different adapters or barcodes are used for separating data of specific samples in silico.

A: We now have clarified this: The underlined adapters added in the first PCR are later used in the Nextera library prep to add barcodes and Illumina adapters. We changed it to: “adapters for MiSeq sequencing“ to “adapters for library preparation”.

C: Lines 177-181: Were functions predicted using DNA or RNA data?

A: We now predicted the functions with both, 16S rRNA genes and -transcripts. We have changed “reads of the ASV table were rarefied to” to “ASV reads of the 16S rRNA gens and -transcripts were rarefied to” and display both analyses in figure 5A and B.

C: Line 195-196:  “Overall, the mean diversity and richness from Upp towards Pio3 were higher compared to values of Pio4 to Pio5” ---- This is not true. They are not significantly different according to Fig. S1.

A: We agree and changed the sentence to “The mean diversity from Pio1 towards Pio3 were higher compared to Edge and Mud1-2”

C: Line 199:  change  “was” into  “appeared to be”  

A: We now state: “the mean diversity was the highest” to “the mean diversity appeared to be the highest”

C: Lines 200-201: “probably due to the high heterogeneity between the individual sites of the transect”

A: This comment is not clear to us.

C: Line 227: “A clear separation of samples from more terrestrial versus marine sites…”  Basically, this hypothesis on difference in community structure should be statistically tested using, for example, ANOSIM or PerMANOVA, rather than simply visual perceives.

A: We agree, we have now done a PERMANOVA to test our hypotheses on a statistical level and integrated the values in the text and a new supplementary table (Table S2).

C:  Line 240:  “the fitted geographic distance between the samples in meters. “---- This is apparently incorrect.

A: We agree, we changed it to “fitted geographic distance to the upper salt marsh in meters”

C:  Figure 4. Please label DNA and RNA to the two figures.

A: We extended the figure description. Furthermore, at many positions in the text our terms “16S rRNA gene” and “16S rRNA transcript” were changed or extended by DNA- and RNA based analyses, respectively, for clarification.

C: In the section of “3.5. Functional gene prediction resolves zonal variations”, trends of functions are generally described, but not statistically tested, making the conclusions weak.

A: That is true. However, as we do not have replicates for each sampling site, a statistical test would not be valid.

C: The discussion should be greatly simplified, as many points only are reviewing previous work (thus do not really matter) rather than focusing on new findings of this work. Also, similar studies and existing knowledges on the tidal zonation effects on bacterial diversity and functions in other plants-dominated coastal ecosystems (e.g., mangrove) may be compared and discussed.

A: We have now restructured the discussion part by shortening and integrating subheadings. However, we do think, that a comparison with other findings have to be part of the discussion. We do not understand the question about the mangroves. We do discuss the work of Chuvochina, et al (2021): …” They sampled the North Stradbroke Island in Queensland (Australia) along a salinity gradient from freshwater wetlands via salt marshes to mangrove sediments. Here, Alphaproteobacteria were most abundant in the freshwater environment, whereas the salt marshes showed an increase in Actinobacteria, Chlorobi, Gamma- and Deltaproteobacteria, with the latter showing higher abundances in the mangroves.”

Reviewer 2 Report

Dear Authors,

I found you manuscript titled : “ Seasonal and zonal succession of bacterial communities in North Sea salt marsh sediments” as an interesting. It is well written and the results are well documented.

I only suggest that increase of figures quality. Figure 4. The quality of legend in this figure is hardly readable. Also at Figure 5 font is  too small.

Author Response

Comment: I found you manuscript titled : “ Seasonal and zonal succession of bacterial communities in North Sea salt marsh sediments” as an interesting. It is well written and the results are well documented.

Answer: We appreciate that reviewer 2 likes our study. We hope, that our changes suggested by all reviewers has improved the manuscript even more.

C: I only suggest that increase of figures quality. Figure 4. The quality of legend in this figure is hardly readable. Also at Figure 5 font is  too small.

A: We agree and have increased the figure qualities and the fonds in figures 4 and 5.

Reviewer 3 Report

In the present work entitled “Seasonal and zonal succession of bacterial communities in North Sea brackish marsh sediments", the authors make an interesting analysis of microbial biodiversity changing in a transept along the land-sea transition zone of the German island of Spiekeroog in the North Sea on a seasonal scale.

Using both rDNA that rRNA (transcription) 16s amplicon biodiversity analysis and gene prediction approach to infer the abundance of the gene involved in nitrogen and sulfur cycles.

The article is well organized and written but there are some decisions made by the authors that are very difficult for me to understand:

First of all, why did the authors decide to exclude Archaea from all the analysis?

There is a myriad of articles highlighting the importance of archaea in the marine and transitional environment involved in the two cycles (nitrogen and sulfur) that the authors would like to describe in this article.

Why decide a priori not to treat these organisms given their great importance?

I am aware that often basic 16s amplicon approach with universal primers tends to underestimate the presence of archaea even in these environments, in favor of bacteria.

Especially for the nitrogen and sulfur cycle in sediments with poor oxygenation where both oxygenation and reduction of nitrogen and sulfur are often mediated by archaea as well as by bacteria. This decision in my opinion risks involving all the results presented here.

Furthermore, using ASV, why did the authors limit themselves to doing all the analyzes only at the level of the Order? The authors could have gone into more detail trying to define the genera or species of bacteria (known from the genomic point of view) to give more strength to the proposed theories.

Although the authors used interesting statistical approaches, it is unclear why the authors did not use the Unifrac distance metric to link bacterial biodiversity to abiotic parameters. Compared to the Bray-Curtis Unifrac dissimilarities, especially if weighed, it could give more information than the Bray-Curtis dissimilarities.

Author Response

Comment: In the present work entitled “Seasonal and zonal succession of bacterial communities in North Sea brackish marsh sediments", the authors make an interesting analysis of microbial biodiversity changing in a transept along the land-sea transition zone of the German island of Spiekeroog in the North Sea on a seasonal scale. Using both rDNA that rRNA (transcription) 16s amplicon biodiversity analysis and gene prediction approach to infer the abundance of the gene involved in nitrogen and sulfur cycles. The article is well organized and written but there are some decisions made by the authors that are very difficult for me to understand. 

Answer: Thank you for your interest in our study. We hope, our changes in the revised version will clarify your concernes. 

C: First of all, why did the authors decide to exclude Archaea from all the analysis? There is a myriad of articles highlighting the importance of archaea in the marine and transitional environment involved in the two cycles (nitrogen and sulfur) that the authors would like to describe in this article. Why decide a priori not to treat these organisms given their great importance? I am aware that often basic 16s amplicon approach with universal primers tends to underestimate the presence of archaea even in these environments, in favor of bacteria. Especially for the nitrogen and sulfur cycle in sediments with poor oxygenation where both oxygenation and reduction of nitrogen and sulfur are often mediated by archaea as well as by bacteria. This decision in my opinion risks involving all the results presented here.

 A: We agree and are aware of this fact. Archaea are an important component of microbial communities and are involved in many metabolic processes. The primers used in our study are discriminating archaea and these organisms are thus not included in our study. Especially the role of ammonia oxidising archaea is very important in marine ecosystems. Thus, our results might underestimate the potential of the benthic microbial community for nitrification. We have now stated this in the discussion. However, any environmental study always provides a limited observation window. For instance, we also excluded eukaryotic primary producers like micro phyto benthos, or other heterotrophic eukaryotes. Both groups of organisms are definitely also involved in elemental cycling within the investigated salt marshes.

C: Furthermore, using ASV, why did the authors limit themselves to doing all the analyzes only at the level of the Order? The authors could have gone into more detail trying to define the genera or species of bacteria (known from the genomic point of view) to give more strength to the proposed theories.

A: For overviewing the bacterial community structures, we decided to limit ourselves to order level as a more detailed taxonomic resolution would have been too complex to display in a bar chart (Figure 4). However, all other analyses including alpha diversity measures (Figure S1), distance decay (Figure 2), NMDS (Figure 3) and tax4fun (Figure 5) are based on ASVs. Especially the tax4fun approach takes functional predictions on genomic level into account.

C: Although the authors used interesting statistical approaches, it is unclear why the authors did not use the Unifrac distance metric to link bacterial biodiversity to abiotic parameters. Compared to the Bray-Curtis Unifrac dissimilarities, especially if weighed, it could give more information than the Bray-Curtis dissimilarities.

A: We agree, that abiotic parameters are mostly lacking and that these would be interesting to incorporate into the betadiversity. The only parameters that were measured during the campaigns were e.g. water content, porewater salinity, temperature or atmospheric pressure. Additionally, these parameters were only measured for a subset of samples and not for all sites along the transect. Thus, we can not incorporate a meaningful set of environmental data relevant into our analysis (e.g. concentrations of nitrate, nitrite, ammonium, sulfate, …).

We agree, that Unifrac distance metric can be advantageous in some aspects. This is especially the case when phylogenomic (dis-)similarities should be considered. However, here we used the Bray-Curtis dissimilarity, as we consider it to be well suited to display differences originating from niche occupation rather than evolutional separation.

Round 2

Reviewer 1 Report

The authors have addressed all my concerns.